# The ATF3–OPG Axis Contributes to Bone Formation by Regulating the Differentiation of Osteoclasts, Osteoblasts, and Adipocytes

**DOI:** 10.3390/ijms23073500

**Published:** 2022-03-23

**Authors:** Jung Ha Kim, Kabsun Kim, Inyoung Kim, Semun Seong, Jeong-Tae Koh, Nacksung Kim

**Affiliations:** 1Department of Pharmacology, Chonnam National University Medical School, Gwangju 61469, Korea; kjhpw@hanmail.net (J.H.K.); kabsun@hanmail.net (K.K.); doll517@naver.com (I.K.); iamsemun@chonnam.edu (S.S.); 2Hard-Tissue Biointerface Research Center, School of Dentistry, Chonnam National University, Gwangju 61186, Korea; jtkoh@chonnam.ac.kr; 3Department of Pharmacology and Dental Therapeutics, School of Dentistry, Chonnam National University, Gwangju 61186, Korea

**Keywords:** ATF3, osteoblast, osteoclast, adipocyte, OPG

## Abstract

Activating transcription factor 3 (ATF3) has been identified as a negative regulator of osteoblast differentiation in in vitro study. However, it was not associated with osteoblast differentiation in in vivo study. To provide an understanding of the discrepancy between the in vivo and in vitro findings regarding the function of ATF3 in osteoblasts, we investigated the unidentified roles of ATF3 in osteoblast biology. ATF3 enhanced osteoprotegerin (OPG) production, not only in osteoblast precursor cells, but also during osteoblast differentiation and osteoblastic adipocyte differentiation. In addition, ATF3 increased nodule formation in immature osteoblasts and decreased osteoblast-dependent osteoclast formation, as well as the transdifferentiation of osteoblasts to adipocytes. However, all these effects were reversed by the OPG neutralizing antibody. Taken together, these results suggest that ATF3 contributes to bone homeostasis by regulating the differentiation of various cell types in the bone microenvironment, including osteoblasts, osteoclasts, and adipocytes via inducing OPG production.

## 1. Introduction

Osteoblasts are one of the major cell types that regulate bone homeostasis and are derived from mesenchymal stem cells (MSCs) [1,2]. In addition to their primary function of bone formation, osteoblasts contribute to maintaining bone homeostasis by the indirect regulation of bone resorption by osteoclasts [3]. Osteoblast differentiation is controlled by various cytokines, such as bone morphogenetic proteins (BMPs), transforming growth factor β, and Wnt. In particular, BMPs are the most potent osteogenic factors that stimulate the differentiation of mesenchymal cells into osteoblasts, inducing bone formation [4,5,6,7]. BMPs transduce signals by binding with complexes of type I and II serine/threonine kinase receptors to activate Smad signaling and regulate the transcription of osteoblast-related transcription factors, such as the runt-related gene 2 (Runx2), alkaline phosphatase (ALP), bone sialoprotein (BSP), and osteocalcin [7,8,9,10].

Since the imbalance between bone formation and bone resorption causes bone diseases, such as osteoporosis, bone formation and bone resorption are coupled via complex mechanisms to prevent the imbalance and to maintain bone homeostasis [11,12,13]. As part of coupled signaling between osteoblasts and bone-resorbing osteoclasts, osteoblasts produce the receptor activator of the NF-κB ligand (RANKL) and its decoy receptor, osteoprotegerin (OPG), to support and attenuate osteoclast differentiation, respectively [3,14]. RANKL induces osteoclast differentiation and bone resorption by binding to its receptor, RANK, which transduces signals by recruiting adaptor molecules on the surface of osteoclast precursor cells [15,16,17]. On the contrary, OPG inhibits osteoclast differentiation by preventing RANKL from binding to RANK. Hence, osteoblasts can indirectly control osteoclast differentiation and thereby prevent excessive osteoclast-mediated bone loss by regulating the ratio of RANKL to OPG [3].

MSCs also have the ability to differentiate into adipocytes, another cell that contributes to bone homeostasis [18,19]. The increase in bone marrow adipose tissue in senile patients with osteoporosis suggests a strong association between the increase in adipocytes and bone loss [20,21,22]. The tendency of MSCs to differentiate into adipocytes at the expense of osteoblasts, or the transdifferentiation of osteoblasts into adipocytes, causes bone loss accompanied by excessive accumulation of bone marrow adipocytes [23].

Activating transcription factor 3 (ATF3) belongs to the ATF/cAMP-responsive element-binding protein (CREB) family and is involved in the pathogenesis of various diseases, such as cancer, atherosclerosis, infections, and hypospadias [24]. Similar to other ATF/CREB transcription factors, ATF3 binds to the canonical ATF/CRE cis-regulatory element, or the similar AP-1 site, via its basic leucine zipper domain (bZIP) to regulate the mRNA expression of various target genes [25,26]. Recent studies have shown that ATF3 may play an important role in bone metabolism by regulating osteoclast and osteoblast differentiation [24,27,28]. Both in vivo and in vitro studies confirmed that ATF3 is a positive regulator of osteoclast differentiation. Among the ATF3/CREB family members, ATF4 and CREB are positive regulators of osteoblast differentiation, whereas ATF3 is a negative regulator of osteoblast differentiation, since it suppresses ALP expression and activity in the in vitro study; however, it was not associated with osteoblast differentiation in the in vivo study [29,30]. In the current study, we further investigated the role of ATF3 in osteoblasts, to understand the discrepancy between in vivo and in vitro findings regarding the function of ATF3 in osteoblasts.

We discovered a novel function of ATF3: Inducing OPG expression in osteoblasts. Osteoblastic ATF3 is involved in the differentiation of osteoclasts, osteoblasts, and adipocytes via OPG production. Although ATF3 inhibits osteoblast differentiation by suppressing *Alpl* expression, ATF3-induced OPG production may enhance bone formation by increasing osteoblast differentiation and decreasing adipocyte differentiation. In addition, ATF3 may prevent excessive osteoclast differentiation by producing OPG. Our findings collectively suggest that osteoblastic ATF3 contributes to the maintenance of bone homeostasis through a known function of the suppression of ALP expression and activity, and a novel function of regulating the coupled signaling between osteoblasts and osteoclasts via OPG production.

## 2. Results

### 2.1. ATF3 Inhibits Osteoblast Differentiation and Bone Formation in Primary Osteoblasts

It has been previously reported that ATF3 inhibits osteoblast differentiation by downregulating ALP expression and activation in the MC3T3-E1 osteoblast-like cell line [28]. To confirm the results of the previous report, we tested whether the overexpression of ATF3 inhibits osteoblast differentiation and bone formation in primary osteoblasts. ATF3 overexpression blocked ALP activation and nodule formation (Figure 1a,b). Real-time PCR results revealed that ATF3 overexpression significantly reduced the mRNA expression of osteogenic genes, including *Alpl*, *Ibsp*, and *Bglap*, except *Runx2* (Figure 1c). These results indicated that ATF3 negatively regulates osteoblast differentiation and function in primary osteoblasts by downregulating ALP expression and activation, as observed previously in the MC3T3-E1 osteoblast-like cell line.

### 2.2. ATF3 Induces the Expression of Tnfrsf11b (Opg)

Next, to test whether ATF3 contributes to coupling signals between osteoclasts and osteoblasts, we examined the effects of ATF3 on the expression of *Tnfsf11* (*Rankl*) and *Opg*, which are produced from osteoblasts and are involved in the regulation of osteoclastogenesis. The overexpression of ATF3 in osteoblasts induced the mRNA expression of *Opg* without affecting *Rankl* mRNA levels, regardless of the presence or absence of 1,25(OH)_2_ vitamin D_3_ (Vit D_3_) (Figure 2a). Conversely, the downregulation of ATF3 significantly reduced *Opg* expression (Figure 2b). In addition, in the co-culture experiments, it was observed that the overexpression or downregulation of ATF3 in osteoblasts decreased or increased the ability of osteoblasts to support osteoclastogenesis, respectively (Figure 2c,d). The ATF3-mediated inhibition of osteoclast formation was restored when cells were treated with RANKL exogenously (Figure 2e). These results suggested that osteoblastic ATF3 is involved in the regulation of osteoclastogenesis through inducing OPG production.

### 2.3. The Net Effect of ATF3 May Be Biased towards Inhibition of Osteoclastogenesis

We previously found that ATF3 positively regulates RANKL-induced osteoclast differentiation in osteoclast precursor cells [27]. However, in the present study, we demonstrated that osteoblastic ATF3 indirectly inhibits osteoclast differentiation by enhancing *Opg* expression. To further clarify the effect of ATF3 on osteoclast differentiation, osteoblasts and osteoclast precursor cells were transduced with the control or ATF3 retrovirus, and then co-cultured in the presence of Vit D_3_ and prostaglandin E2 (PGE_2_), as shown in Figure 3. As expected, the overexpression of ATF3 in osteoclast precursor cells showed an increase in osteoclast formation, and the overexpression of ATF3 in osteoblasts exhibited a decrease in osteoclast formation. More importantly, the overexpression of ATF3 in both osteoclast precursor cells and osteoblasts showed significantly reduced osteoclast formation compared to the control (Figure 3). These results are due to the anti-osteoclastogenic effects of osteoblastic ATF3 that surpass the direct stimulatory effects of ATF3 on RANKL-induced osteoclast differentiation in osteoclast precursor cells. These results suggested that the net balance between the direct pro-osteoclastogenic effect and the indirect anti-osteoclastogenic effect of ATF3 was biased towards inhibiting osteoclastogenesis.

### 2.4. OPG Promotes Osteoblast Differentiation and Attenuates Transdifferentiation of Osteoblasts to Adipocytes

To further investigate the mechanistic role of ATF3 through OPG induction in osteoblasts, we first examined the effects of OPG on osteoblast differentiation and the transdifferentiation of osteoblasts to adipocytes. OPG overexpression enhanced BMP2-induced ALP activation and nodule formation, as well as increasing the mRNA expression of osteoblastogenic genes, such as *Alpl*, *Ibsp*, and *Bglap* (Figure 4a–c). When OPG-transduced osteoblasts were cultured in an adipogenic medium, including insulin, dexamethasone, IBMX, and rosiglitazone, the formation of lipid droplets was significantly reduced compared to control osteoblasts (Figure 4d). In addition, OPG overexpression in osteoblasts significantly suppressed the expression of adipocyte marker genes, including *Cebpα*, *Pparγ*, and *Fabp4* compared to the control cells when osteoblasts were cultured in an adipogenic medium (Figure 4e). These results suggest that OPG positively and negatively regulates osteoblast differentiation and the transdifferentiation of osteoblasts to adipocytes, respectively.

### 2.5. ATF3 Acts like OPG in the Differentiation of Osteoblasts and the Transdifferentiation of Osteoblasts to Adipocytes

Since a difference in the function of ATF3 and OPG in osteoblast differentiation was observed, we tested whether ATF3 had a stage-dependent function in regulating osteoblast differentiation. ATF3 was overexpressed before, or on the fourth day after, the induction of osteoblast differentiation, and the cells were cultured for 6 or 3 days in an osteogenic medium, respectively. ATF3 overexpression strongly inhibited nodule formation in the former, whereas it slightly increased it in the latter (Figure 5a). When ATF3 was overexpressed on the fourth day after the induction of osteoblast differentiation, the suppressive effect of osteogenic genes, which was observed when ATF3 was overexpressed before the induction of osteoblast differentiation, was slightly alleviated or reversed (Figure 5b). The transdifferentiation of osteoblasts to adipocytes was reduced upon the overexpression of ATF3 (Figure 5c). Consistent with these results, the expression of adipocyte marker genes, such as *Cebpα*, *Pparγ*, and *Fabp4* was significantly suppressed by ATF3 overexpression (Figure 5d). Collectively, these results suggested that ATF3 could enhance bone formation at the late stage of osteoblast differentiation and attenuate lipid droplet formation.

### 2.6. ATF3 Is Involved in Controlling the Differentiation of Osteoclasts, Osteoblasts, and Adipocytes through OPG Induction

Since we confirmed the induction of *Opg* by ATF3 during osteoblast differentiation as well as adipocyte differentiation (Figure 6a), we further investigated how ATF3 regulates *Opg* transcription. We conducted a chromatin immunoprecipitation (ChIP) analysis to examine whether ATF3 binds to candidate ATF3-binding motifs within the *Opg* promoter regions (Figure 6b). The ChIP analysis revealed that ATF3 binds to one of the two candidate binding sites, indicating that ATF3 induced *Opg* transcription via direct binding to an *Opg* promoter (Figure 6b). Next, we examined whether ATF3 regulated the differentiation of osteoclasts, osteoblasts, and adipocytes through *Opg* induction. In the co-culture experiments, the inhibition of osteoblast-dependent osteoclast differentiation by ATF3 overexpression was recovered by the addition of the OPG neutralizing antibody (Figure 6c). Likewise, OPG neutralization significantly reversed the effect of ATF3 in regulating not only bone formation, but also lipid droplet formation (Figure 6d,e). These results suggest that ATF3 induces *Opg* expression to regulate osteoblast-dependent osteoclast differentiation, osteoblast differentiation at the late stage, and the transdifferentiation of osteoblasts to adipocytes.

## 3. Discussion

ATF4 and CREB, which belong to the ATF/CREB family along with ATF3, are known to be positive regulators of osteoblast differentiation. However, the role of ATF3 in osteoblasts still remains controversial [29,30]. ATF3 is known to prevent osteoblast differentiation by specifically suppressing *Alpl* expression in the MC3T3-E1 osteoblast-like cell line, and our results confirmed that ATF3 inhibits osteoblast differentiation in primary osteoblasts as well [28]. However, an in vivo study suggested that ATF3 may not be involved in osteoblast differentiation, since no abnormalities were observed in mice with the conditional deletion of ATF3 in osteoblasts under both physiological and pathological conditions [24]. This discrepancy between the findings obtained from the in vivo and in vitro studies may be due to some unidentified role of ATF3 in osteoblasts. In the present study, we found a novel role of ATF3 in inducing OPG production in both unstimulated and Vit D_3_ stimulated osteoblasts. As expected, ATF3 inhibited osteoblast-dependent osteoclast differentiation, and this effect was strong enough to surpass the pro-osteoclastogenic effect of ATF3 in osteoclast precursor cells. OPG production by osteoblasts is critical for blocking excessive bone resorption and coupling bone formation and resorption [3]. Thus, the deficiency of ATF3 in osteoblasts leads to an increase in osteoblast differentiation and a decrease in OPG production, which possibly exhibits no abnormalities in vivo.

The p38α–CREB axis has been shown to regulate *Opg* transcription in bone marrow-derived MSCs (BMMSCs) [31]. When we tested whether the p38α–CREB axis regulates *Opg* transcription in osteoblasts as well, CREB overexpression in osteoblasts did not affect *Opg* transcription, while the inhibition of the p38 pathway suppressed *Opg* transcription in Vit D_3_-stimulated osteoblasts (Appendix A). These results indicate that ATF3 is more important than CREB for *Opg* transcription in osteoblasts. Previously, it has been reported that ATF3 induction is mediated by p38 and c-Jun N-terminal kinase in *Streptococcus pneumoniae*-infected RAW 264.7 cells [32]. Therefore, we tested whether the p38α–ATF3 axis regulates *Opg* transcription in osteoblasts. Unexpectedly, treatment with a p38 inhibitor did not abolish the expression of endogenous or overexpressed ATF3. Furthermore, the reduced expression of *Opg* upon p38 inhibition was only slightly recovered with ATF3 overexpression (Appendix A). Interestingly, according to previous studies, the ablation of p38α in osteoblasts failed to affect osteoclast differentiation and bone resorption, suggesting that p38α-depleted osteoblasts may have redundant pathways for the p38α–OPG axis to regulate *Opg* transcription [33,34]. Taken together, these results suggest that ATF3 may be one of the alternative factors compensating for the p38α–OPG axis to regulate *Opg* transcription in p38α-depleted osteoblasts, and ATF3 and p38α regulate *O**pg* transcription independently of each other. However, in vitro results showed reduced *O**pg* transcription upon treatment with a p38 inhibitor. This effect was not reversed upon ATF3 overexpression and needs further investigation. Therefore, it is necessary to clarify whether any unexpected non-specific activities of the p38 inhibitor regulate ATF3 activation.

ATF3 binds to the canonical ATF/CRE *cis*-regulatory element (5′-TGACGTCA-3′) or the similar AP-1 site (5′-TGA(C/G)TCA-3′) to regulate the transcription of numerous target genes [35]. The ChIP analysis showed that *Opg* is an ATF3-regulated gene whose expression is controlled by the direct binding of ATF3 to the ATF3 motif localized in the promoter. As the *Opg* promoter region also contains AP-1 sites, it is still possible that ATF3 binds to the AP-1 sites to regulate *Opg* transcription. Furthermore, ATF3 could regulate gene expression by binding to the distal *cis*-regulatory element localized in active enhancers enriched with p300 and H3K27ac, instead of binding to the proximal ATF3-binding motif [36]. Thus, further studies are required to elucidate the regulatory mechanism of ATF3 in *Opg* transcription.

OPG knockout mice exhibit porous, but thick, cortical bone due to rapid bone remodeling [37]. Although the obvious bone phenotype of OPG knockout mice suggests that OPG plays an important role in the inhibition of bone resorption and in coupled signaling between osteoclasts to osteoblasts, the functional potential of OPG on bone formation itself cannot be completely excluded. The osteogenic differentiation abilities of BMMSCs derived from OPG knockout cells are weaker than those of BMMSCs derived from wild-type cells in vitro, suggesting that OPG regulates bone formation in a cell-autonomous manner [37]. Consistent with these results, our findings revealed that OPG positively regulates osteoblast differentiation. Although ATF3 strongly increased *Opg* expression during osteoblast differentiation, the overexpression of ATF3 inhibited osteoblast differentiation via suppressing *Alpl* expression in contrast to the overexpression of OPG. To minimize the effect of the ATF3-mediated suppression of *Alpl*, ATF3 was overexpressed in immature osteoblasts in which *Alpl* expression was sufficiently induced, and ATF3 increased osteoblast differentiation by inducing *Opg* expression. ATF3 appears to have dual roles as a negative regulator and a positive regulator of osteoblast differentiation, depending on the differentiation stage of osteoblasts. Interestingly, Choe et al. reported that ATF3 induces calcification in vascular smooth muscle cells, which somewhat contradicts the osteogenesis-preventing property of ATF3 in the MC3T3-E1 osteoblast-like cell line. This could perhaps be due to a cell type-specific effect or the bidirectional transcriptional activity of ATF3 [38]. Therefore, ATF3 can negatively or positively regulate osteoblast differentiation depending on whether the major target gene is *Alpl* or *Opg*, which may be one reason that osteoblast-specific ATF3 knockout mice exhibited no apparent change in osteoblastic parameters.

An increased number of adipocytes were observed in the bone marrow of OPG knockout mice, compared to control mice [37]. In vitro studies also showed that OPG-deficient BMMSCs could be differentiated into adipocytes more rapidly than wild-type BMMSCs, but they were more difficult to differentiate into osteoblasts [37]. As expected, we confirmed that the overexpression of OPG in osteoblasts inhibited the transdifferentiation of osteoblasts to adipocytes. In the present study, we showed that ATF3 suppressed the transdifferentiation of osteoblasts to adipocytes, similar to OPG, and this was associated with OPG production. ATF3 is known to repress adipocyte differentiation by inhibiting the expression of *Cebpα*, as well as the expression and transactivation of *Pparγ* [39,40]. Considering that CEBPα and PPARγ are critical regulators of adipocyte differentiation, the inhibitory effect of ATF3 on the adipogenic transdifferentiation of osteoblasts may rely on the inhibition of *Cebpa* and *Pparγ* expressions. However, since ATF3 induces *Opg* expression during adipogenic transdifferentiation, and the OPG neutralizing antibody abolished the inhibitory effect of ATF3, it can be deduced that ATF3 contributes to the inhibition of adipogenic transdifferentiation, in part, by upregulating *Opg* expression.

In summary, in the present study, we demonstrated that osteoblastic ATF3 is involved in bone metabolism in a variety of ways to maintain bone homeostasis. Although ATF3 is known as an anti-osteogenic and pro-osteoclastogenic factor, the ability of osteoblastic ATF3 to induce *Opg* expression may contribute to bone formation by inhibiting osteoclast differentiation, increasing osteoblast differentiation, and inhibiting adipocyte differentiation.

## 4. Materials and Methods

### 4.1. Reagents

The recombinant human BMP2 protein was purchased from Cowellmedi (Busan, Korea). Ascorbic acid was purchased from Junsei Chemical (Nihonbashi-Honcho, Japan). β-glycerophosphate, insulin, rosiglitazone, dexamethasone, and 3-isobutyl-1-methylanthine (IBMX) were purchased from Sigma-Aldrich (St. Louis, MO, USA).

### 4.2. Osteoblast Differentiation

All experiments using cells obtained from mice were approved by the Chonnam National University Medical School Research Institutional Animal Care and Use Committee (CNU IACUC-H-2020-26). Primary osteoblast precursor cells were isolated from the calvaria of a neonatal mouse. The cells were extracted from the calvaria using enzymes containing 0.1% collagenase (Life Technologies, Carlsbad, CA, USA) and 0.2% dispase II (Roche Diagnostics GmbH, Mannheim, Germany) for 10 min each, at 37 °C, four times. For the differentiation experiments, cells were plated on a 6-well culture dish or a 48-well culture dish at a density of 3 × 10^5^ cells or 3 × 10^4^, respectively, and then cells were cultured in osteogenic medium (OGM) containing BMP2 (100 ng/mL) (Cowellmedi, Busan, Korea), ascorbic acid (50 µg/mL) (Junsei Chemical, Nihonbashi-Honcho, Japan), and β-glycerophosphate (100 mM) (Sigma-Aldrich, St. Louis, MO, USA). Cells cultured for 3 days were lysed using a lysis buffer (50 mM Tris-HCl (pH 7.4), 1% Triton X-100, 150 mM NaCl, and 1 mM EDTA) to measure ALP activity. Cell lysates were incubated with a *p*-nitrophenyl phosphate substrate (Sigma-Aldrich, St. Louis, MO, USA), and ALP activity was analyzed by measuring the absorbance at 405 nm using a spectrophotometer. Cells cultured for 6 days were fixed with 3.7% formalin for 10 min, followed by the addition of Alizarin Red (40 mM, pH 4.2) for 10 min. Stained cells were rinsed with PBS to remove nonspecific staining and visualized using the CanoScan 9000F image scanner (Canon Inc., Tokyo, Japan). To quantify the Alizarin Red staining, 10% acetic acid was added and incubated for 30 min at room temperature. The absorbance of the eluent was measured at 405 nm using a spectrophotometer.

### 4.3. Osteoclast Differentiation

Wild-type ICR mice were euthanized according to Chonnam National University Medical School Research Institutional Animal Care and Use Committee, followed by the isolation of femurs and tibias. The long bones were flushed with α-MEM to obtain bone marrow cells. The isolated cells were incubated with a Red Blood Cell Lysis Buffer (Sigma-Aldrich, St. Louis, MO, USA) to remove the red blood cells. The remaining cells were used as osteoclast precursor cells and co-cultured with calvaria-derived primary osteoblast precursors in the presence of 1,25(OH)_2_ vitamin D_3_ (10 nM) for 5 days. Subsequently, the culture medium was replaced every 3 days. The cultured cells were fixed with 3.7% formalin and a TRAP stain was conducted for the determination of TRAP-positive multinuclear cells. TRAP-positive cells with three or more nuclei were determined as multinuclear osteoclasts and the number was quantified. The stained cells were observed using the Leica DMIRB microscope equipped with an N Plan 10× 0.25 numerical aperture objective lens (Leica, Wetzlar, Germany). Images were captured using the ProgRes CapturePro software, 2.7.7 (13 January 2010), (Jenoptik, Jena, Germany).

### 4.4. Adipocyte Differentiation

Primary osteoblast precursor cells were induced to adipocytes by culturing in an adipogenic medium in the presence of insulin (10 µg/mL), dexamethasone (0.1 µM), rosiglitazone (1 µM), and IBMX (0.5 mM) for 3 days. Subsequently, the cells were further cultured in an adipogenic medium containing only insulin (10 µg/mL) for 3 days. The fluctuation of the presence or absence of dexamethasone, rosiglitazone, and IBMX was provided every 6 days. The cells were fixed with 3.7% formalin and stained with Oil Red O stain solution (Sigma-Aldrich, St. Louis, MO, USA) for the determination of lipid droplet-containing adipocytes.

### 4.5. Retroviral Gene Transduction

For retroviral infection, Plat E cells were seeded on a 10 cm culture dish at a density of 15 × 10^5^ cells. The next day, the cells were transfected with pMX-FIG (control) or pMX-ATF3 (ATF3) using FuGENE 6 (Promega Corporation, Madison, WI, USA) according to the manufacturer’s instructions. After 48 h, the supernatant was collected and used as a transduction medium after the addition of polybrene (Sigma-Aldrich, St. Louis, MO, USA). Osteoblasts were transduced for 6 h; the medium was then replaced with a growth medium or a differentiation medium.

### 4.6. siRNA Transfection

For siRNA transfection, osteoblasts were transfected with control siRNA or ATF3 siRNA (Dharmacon, Lafayette, CO, USA) using Lipofectamine RNAiMAX (Invitrogen, Carlsbad, CA, USA) according to the manufacturer’s instructions. The medium was replaced with a growth medium or a differentiation medium 4 h after transfection.

### 4.7. OPG Neutralization

To determine the effects of OPG neutralization, cells were cultured with 20 ng/mL anti-OPG (R&D Systems, Inc., Minneapolis, MN, USA) during the incubation period.

### 4.8. ChIP

ChIP was performed using an EZ-ChIPTM Kit (MilliporeSigma, Burlington, MA, USA) according to the manufacturer’s instructions. Briefly, ATF3-transduced osteoblasts were cultured in a growth medium. Cultured cells were treated with 10% formaldehyde for 10 min at room temperature. The chromatin was then sheared by sonication. Protein/DNA complexes were immunoprecipitated with immunoglobulin G or an anti-ATF3 antibody (Abcam, Cambridge, MA, USA). Subsequently, protein/DNA complexes were eluted and subjected to reverse crosslinking, and DNA was purified using the spin columns that were provided with the EZ-ChIPTM Kit (MilliporeSigma, Burlington, MA, USA). Purified DNA was amplified using PCR. The sequences of the primers containing the ATF3-binding sites in the *Opg* promoter region were as follows: +87 to +236 site, 5′-AGG TTT GTC CAG ACA GAG AC-3′ and 5′-GTG GTC CTC GGG AAA CCT CA-3′; -1201 to -1300 site, 5′-TGT AGA TAA TCA ATC TCT C-3′ and 5′-AAA CA TTT TTC TCA AAA TG-3′; -700 to -800 site (negative control), 5′-GTT GCC TAT GGC ATC TTG G-3′ and 5′-GAT TTG CAA AAT AAG GTT C3′.

### 4.9. Real-Time PCR

The total RNA was isolated using the Qiazol reagent (Qiagen, GmbH, Hilden, Germany), and 2 µg of the isolated RNA was reverse transcribed into cDNA using GoScript™ Reverse Transcriptase (Promega Corporation, Madison, WI, USA). Quantitative real-time PCR was performed using the reverse transcribed cDNA in triplicate with Rotor-Gene Q (Qiagen, Hilden, Germany) and SYBR Green (Qiagen, Hilden, Germany). The expression levels of the target genes were normalized against endogenous *Gapdh* levels. The following primers were used: *Gapdh*, 5′-TGA CCA CAG TCC ATG CCA TCA CTG-3′ and 5′-CAG GAG ACA ACC TGG TCC TCA GTG-3′; *Runx2*, 5′-CCC AGC CAC CTT TAC CTA CA-3′ and 5′-CAG CGT CAA CAC CAT CAT TC-3′; *Alpl*, 5′-CAA GGA TAT CGA CGT GAT CAT G-3′ and 5′-GTC AGT CAG GTT GTT CCG ATT C-3′; *Ibsp*, 5′-GGA AGA GGA GAC TTC AAA CGA AG-3′ and 5′-CAT CCA CTT CTG CTT CTT CGT TC-3′; *Bglap*, 5′-ATG AGG ACC CTC TCT CTG CTG CTC AC-3′ and 5′-AGA GCA AAC TGC AGA AGC TGA GAG-3′; *Cebp*α, 5′-AAG AAG TCG GTG GAC AAG AAC AG-3′ and 5′-TGC GCA CCG CGA TGT-3′; *Pparγ*, 5′-TCC AGC ATT TCT GCT CCA CA-3′ and 5′-ACA GAC TCG GCA CTC AAT GG-3′; *Fabp4*, 5′-AAA TCA CCG CAG ACG ACA-3′ and 5′-CAC ATT CCA CCA CCA GCT-3′; *Tnfsf11*, 5′-CCT GAG ACT CCA TGA AAA CGC-3′ and 5′-TCG CTG GGC CAC ATC CAA CCA TGA-3′; *Tnfrsf11b*, 5′-CGG CGT GGT GCA AGC TGG AAC-3′ and 5′-CCT CTT CAC ACA GGG TGA CAT C-3′.

### 4.10. Statistical Analyses

All values are expressed as the mean ± standard deviation (SD). Statistical significance was determined using a two-tailed Student’s *t*-test for two independent samples, or an analysis of variance (ANOVA) with the post-hoc Tukey’s HSD test for multiple group comparisons. Results with *p* < 0.05 were considered statistically significant.

## Figures and Tables

**Figure 1 ijms-23-03500-f001:**
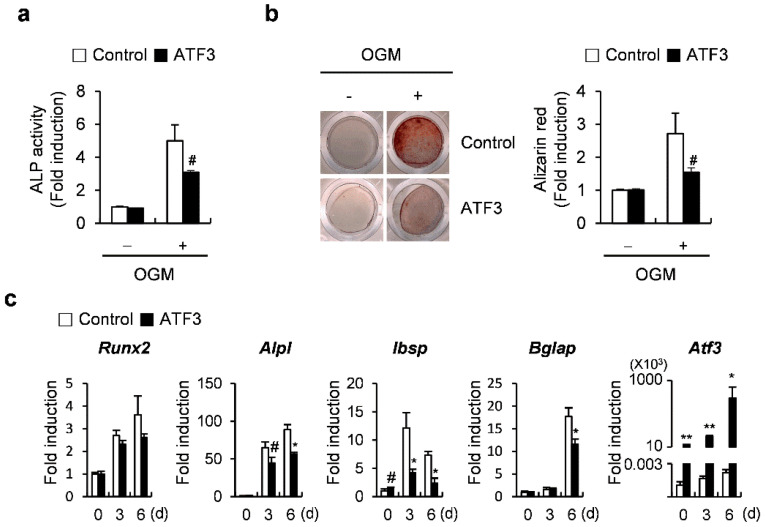
ATF3 attenuates osteoblast differentiation in primary osteoblast precursors. (**a**–**c**) Primary osteoblast precursors transduced with pMX-IRES-EGFP (control) or ATF3 retrovirus were cultured in osteogenic medium (OGM). (**a**) Cells were cultured for 3 days and subjected to ALP assay. (**b**) Cells were cultured for 6 days and stained with Alizarin Red (left panel). Alizarin Red staining was quantified by densitometry at 405 nm (right panel). (**c**) Cells were cultured for the indicated time points and expression levels of the indicated genes were analyzed by real-time PCR. # *p* < 0.05, * *p* < 0.01, ** *p* < 0.001 vs. control.

**Figure 2 ijms-23-03500-f002:**
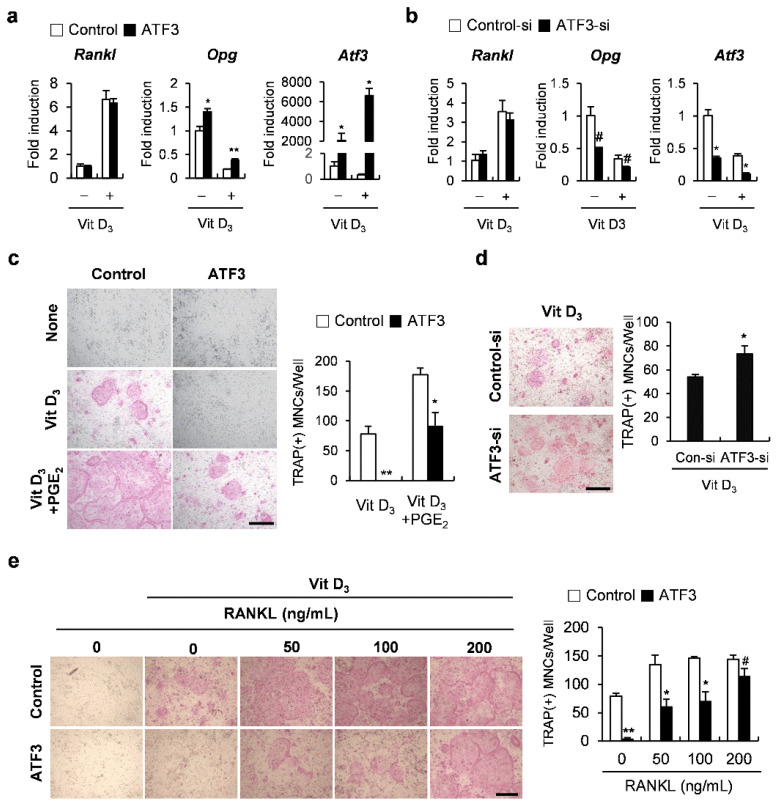
ATF3 augments OPG expression in osteoblasts. (**a**) Primary osteoblast precursors transduced with pMX-IRES-EGFP (control) or ATF3 retrovirus were cultured in the presence or absence of Vit D_3_. The expression levels of the indicated genes were analyzed by real-time PCR. (**b**) Primary osteoblast precursors transfected with negative control siRNA (control-si) or ATF3 siRNA (ATF3-si) were cultured in the presence or absence of Vit D_3_. The expression levels of the indicated genes were analyzed by real-time PCR. (**c**) Primary osteoblast precursors were transduced with pMX-IRES-EGFP (control) or ATF3 retrovirus and were co-cultured with bone marrow cells in the presence or absence of Vit D_3_ and PGE_2_. The cultured cells were stained for TRAP (left panel). TRAP-positive multinucleated cells were quantified (right panel), scale bar: 200 µm. (**d**) Primary osteoblast precursors were transfected with control siRNA (control-si) or ATF3 siRNA (ATF3-si) and were co-cultured with bone marrow cells in the presence or absence of Vit D_3_. The cultured cells were stained for TRAP (left panel). TRAP-positive multinucleated cells were quantified (right panel), scale bar: 200 µm. (**e**) Primary osteoblast precursors were transduced with pMX-IRES-EGFP (control) or ATF3 retrovirus and were co-cultured with bone marrow cells in the presence or absence of Vit D_3_ and with the indicated concentrations of exogenous RANKL. The cultured cells were stained for TRAP (left panel). TRAP-positive multinucleated cells were quantified (right panel), scale bar: 200 µm. # *p* < 0.05, * *p* < 0.01, ** *p* < 0.001 vs. control.

**Figure 3 ijms-23-03500-f003:**
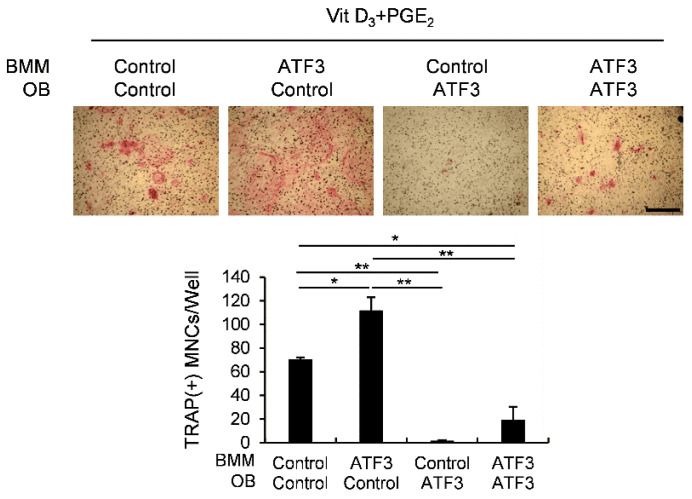
Osteoblastic ATF3 exhibits precedence over pro-osteoclastic function of ATF3 in BMMs. Primary osteoblast precursors and BMMs transduced with pMX-IRES-EGFP (control) or ATF3 retrovirus were co-cultured in the presence of Vit D_3_ and PGE_2_. The cultured cells were stained for TRAP (upper panel). TRAP-positive multinucleated cells were quantified (lower panel), scale bar: 200 µm. * *p* < 0.01, ** *p* < 0.001 vs. control.

**Figure 4 ijms-23-03500-f004:**
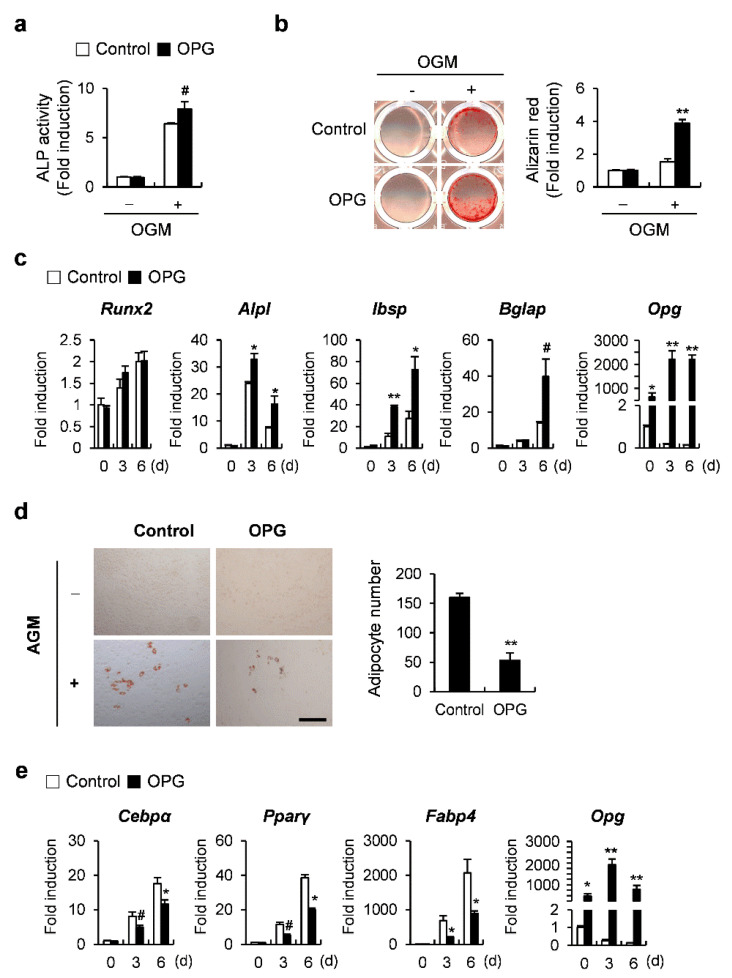
OPG regulates osteoblast differentiation and adipogenic transdifferentiation of primary osteoblast precursors. (**a**–**c**) Primary osteoblast precursors transduced with pMX-IRES-EGFP (control) or OPG retrovirus were cultured in OGM. (**a**) Cells were cultured for 3 days and subjected to ALP assay. (**b**) Cells cultured in OGM for 6 days were stained with Alizarin Red (left panel). Alizarin Red staining was quantified by densitometry at 405 nm (right panel). (**c**) Cells were cultured for the indicated time points and the expression levels of the indicated genes were analyzed by real-time PCR. (**d**,**e**) Primary osteoblast precursors transduced with pMX-IRES-EGFP (control) or OPG retrovirus were cultured in adipogenic medium (AGM). (**d**) Cells cultured in AGM for 6 days were stained with Oil Red O (left panel). Oil Red O-positive lipid droplets containing adipocytes were quantified (right panel), scale bar: 200 µm. (**e**) Cells were cultured for the indicated time points and the expression levels of the indicated genes were analyzed by real-time PCR. # *p* < 0.05, * *p* < 0.01, ** *p* < 0.001 vs. control.

**Figure 5 ijms-23-03500-f005:**
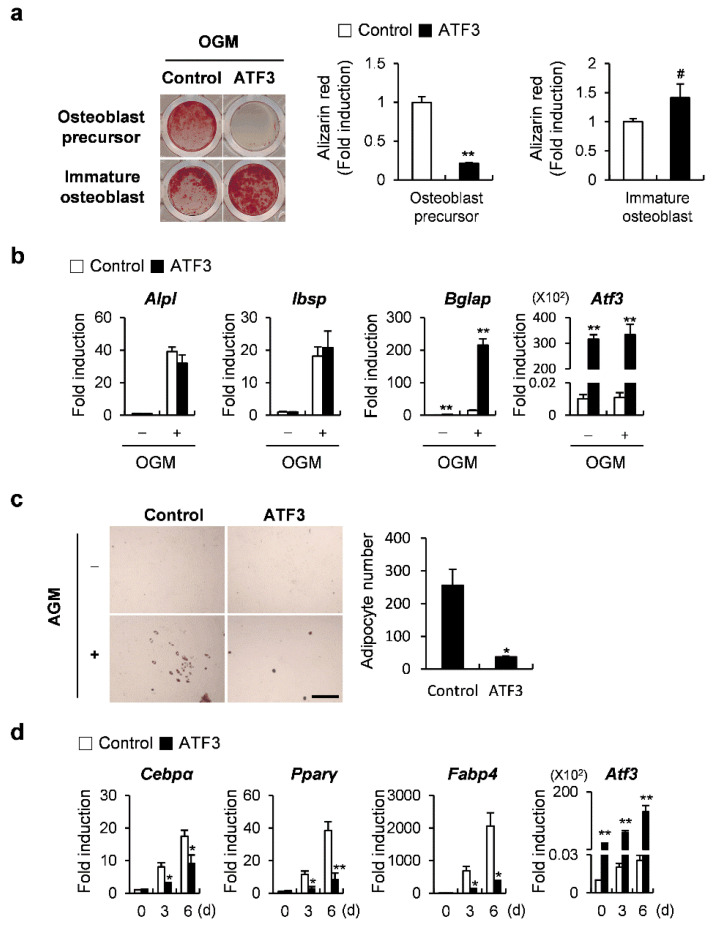
ATF3 regulates osteoblast differentiation and adipogenic transdifferentiation of primary osteoblast precursors similar to OPG. (**a**,**b**) Primary osteoblast precursors transduced with pMX-IRES-EGFP (control) or ATF3 retrovirus at different stages of differentiation were cultured in OGM. (**a**) Cells cultured in OGM for 6 days were stained with Alizarin Red (left panel). Alizarin Red staining was quantified by densitometry at 405 nm (right panel). (**b**) The expression levels of the indicated genes were analyzed by real-time PCR. (**c**,**d**) Primary osteoblast precursors transduced with pMX-IRES-EGFP (control) or ATF3 retrovirus were cultured in AGM. (**c**) Cells cultured in AGM for 6 days were stained with Oil Red O (left panel). Oil Red O-positive lipid droplets containing adipocytes were quantified (right panel), scale bar: 200 µm. (**d**) Cells were cultured for the indicated time points and the expression levels of the indicated genes were analyzed by real-time PCR. # *p* < 0.05, * *p* < 0.01, ** *p* < 0.001 vs. control.

**Figure 6 ijms-23-03500-f006:**
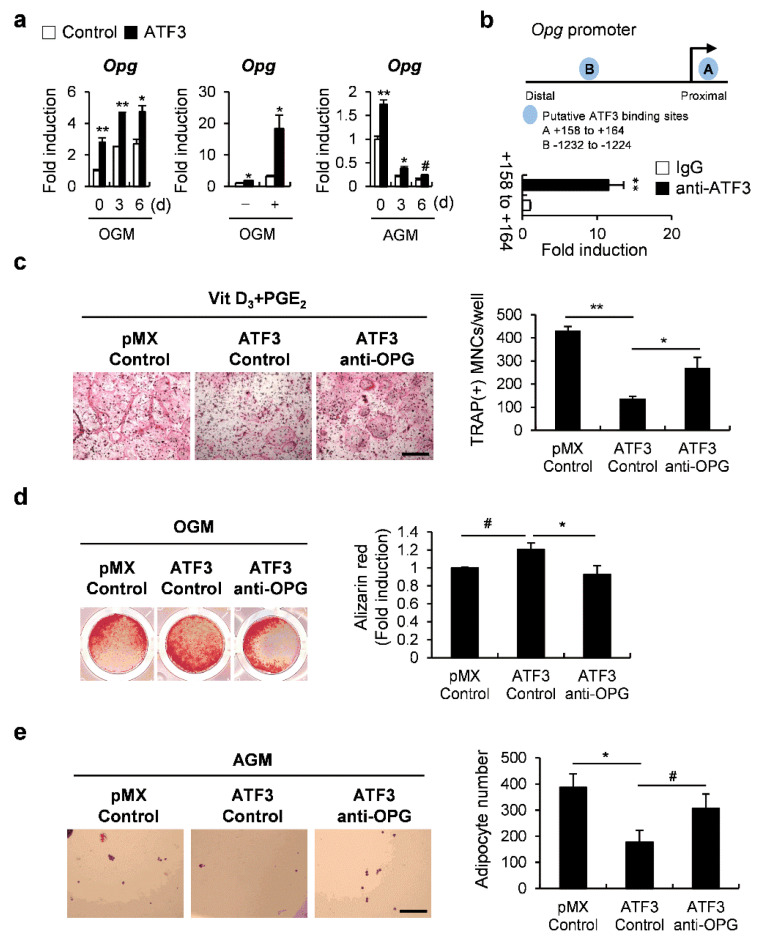
ATF3 simultaneously regulates the differentiation of osteoclasts, osteoblasts, and adipocytes via induction of OPG expression. (**a**) Primary osteoblast precursors transduced with pMX-IRES-EGFP (control) or ATF3 were cultured in OGM (left panel) or AGM (right panel). Cells were cultured for the indicated time points and the expression levels of the indicated genes were analyzed by real-time PCR. (**b**) Illustration of ATF3 candidate binding motifs in an *Opg* promoter (upper panel); chromatin immunoprecipitation assay of osteoblasts was conducted against an ATF3 antibody (lower panel). (**c**) Primary osteoblast precursors transduced with pMX-IRES-EGFP (control) or ATF3 retrovirus were co-cultured with BMCs in the presence of Vit D_3_ and PGE_2_ and in the presence or absence of OPG neutralizing antibody (anti-OPG). The cultured cells were stained for TRAP (left panel). TRAP-positive multinucleated cells were quantified (right panel), scale bar: 200 µm. (**d**) Primary osteoblast precursors were cultured in OGM for 3 days. Immature osteoblasts were transduced with pMX-IRES-EGFP (control) or ATF3 retrovirus and were further cultured in OGM in the presence or absence of anti-OPG. The cultured cells were stained with Alizarin Red (left panel). Alizarin Red staining was quantified by densitometry at 405 nm (right panel). (**e**) Primary osteoblast precursors were transduced with pMX-IRES-EGFP (control) or ATF3 retrovirus in the presence or absence of anti-OPG, and cultured in AGM. The cells cultured in AGM for 6 days were stained with Oil Red O (left panel). Oil Red O-positive lipid droplets containing adipocytes were quantified (right panel), scale bar: 200 µm. # *p* < 0.05, * *p* < 0.01, ** *p* < 0.001 vs. control.

## Data Availability

All data generated or analysed during this study are in this published article.

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
