# Peer review of "The ATF3–OPG Axis Contributes to Bone Formation by Regulating the Differentiation of Osteoclasts, Osteoblasts, and Adipocytes"

_ijms, 2022, doi:10.3390/ijms23073500_

Round 1
Reviewer 1 Report
Major comment
I think this article has two new findings; ATF3 regulates osteoprotegerin (OPG) expression in osteoblasts, and OPG induces osteogenesis in osteoblasts. Clear-cut data support these findings. How do the authors think about the mechanism of action of OPG in osteoblasts? Does OPG work as a decoy receptor of RANKL in osteoblasts? If the osteoblasts have a receptor of RANKL, the additional experiment, the addition of RANKL to the culture of ATF3-or OPG-overexpressed cells, might be worth conducting. Another possibility is that OPG works in an autocrine manner.
Minor comments
- There is no description of methods of retroviral gene transduction and siRNA transfection of ATF into cells.
- Primary osteoblasts and bone marrow cells were obtained from mice in this study. If any committee or council approved this experimental protocol of animal experiments, please describe it.
- Please explain the method of the experiment using an anti-OPG antibody, the manufacture information of the antibody, the duration of antibody treatment.
Reviewer 2 Report
There are several concerns which need address before acceptance. Please find my suggestions below:
- In Figure 1A&B, the authors mentioned “ATF3 overexpression blocked BMP2-induced ALP activation and nodule formation.” Please explain why it is BMP2-induced ALP activity not other osteoblastic factors?
- In this study, ATF3 regulates ALPL and OPG mRNA expression. As a transcription factor, dose ATF3 directly bind to promoter region and regulate the transcriptional activity of ALPL and OPG genes? If so, ChIP assay or luciferase reporter assay must be done. If not, please discuss the possible mechanism(s) in the Discussion section.
- The full meaning of “AGM” should be mentioned in the article.
- Some histograms, such Figure 1C and 2C, the difference between control and ATF3 is too large to be displayed with the same Y-axis coordinate.
- The authors mentioned osteoblastic ATF3 inhibits osteoclastogenesis through inducing OPG production. Dose ATF3 implicate in the treatment of bone loss diseases? Such as osteolytic bone metastasis or osteoporosis. Please discuss it.
Round 2
Reviewer 2 Report
The authors have accordingly addressed all the issues. However, the Methods section has high similarity, it must be revised before acceptance.
Author Response
We edited the Methods section to reduce the high similarity as you pointed. Thank you for your kind comments.